# On the reflectance spectroscopy of snow

Alexander Kokhanovsky(1), Maxim Lamare(2,3), Biagio Di Mauro(4), Ghislain Picard
(2), Laurent Arnaud (2), Marie Dumont (3), François Tuzet (3,2), Carsten Brockmann(5),
Jason E. Box(6)

(1) VITROCISET, Bratustrasse 7, D-64293 Darmstadt, Germany

(2) UGA, CNRS, Institut des Géosciences de l'Environnement (IGE), UMR 5001,

  Grenoble, 38041, France

(3) Meteo-France–CNRS, CNRM UMR 3589, Centre d'Etudes de la Neige, Grenoble,

  France

(4) Department  of Earth and Environmental Sciences, University of Milano-Bicocca, Piazza

  della Scienza, 1 20126 Milan, Italy

(5) Brockmann Consult, Max Planck Strasse 2, Geesthacht, Germany

(6) Geological Survey of Denmark and Greenland (GEUS), Copenhagen, Denmark

**Abstract**

We propose a system of analytical equations to retrieve snow grain size and absorption
coefficient of pollutants from snow reflectance or snow albedo measurements in the visible
and near-infrared regions of the electromagnetic spectrum, where snow single scattering
albedo is close to 1.0. It is assumed that ice grains and impurities (e.g., dust, black and brown
carbon) are externally mixed, the snow layer is semi-infinite and vertically and horizontally
homogeneous. The influence of close–packing effects on reflected light intensity are assumed
to be small and ignored. The system of nonlinear equations is solved analytically in the
assumption that impurities have the spectral absorption coefficient , which obey the Angström
power law, and the impurities  influence the registered spectra only in the visible and not at
near-infrared (and vice versa for ice grains). The theory is validated using spectral reflectance
measurements and albedo of clean and polluted snow at various locations (Antarctica Dome C,
European Alps). The technique to derive the snow albedo (plane and spherical) from

reflectance measurements at a fixed observation geometry is proposed. The technique also enables the simulation of hyperspectral snow reflectance measurements in the broad spectral range from ultraviolet to the near-infrared for a given snow surface in the case, if the actual measurements are performed at restricted number of wavelengths (2-4, depending on the type of snow and the measurement system).

## 1. Introduction

The reflective properties of clean and polluted snow are of importance for various applications including climate (Hansen and Nazarenko, 2007) and environmental pollution (Nazarenko et al., 2017) studies. The spectral snow reflectance is usually studied in the framework of the radiative transfer theory. The application of the numerical methods for the solution of the radiative transfer equation for snow layers has been performed by Mishchenko et al. (1999), Stamnes et al. (2011), and He et al. (2018) among others. The approximate solutions of the radiative transfer equation useful for snow optics and spectroscopy applications have been developed by Warren and Wiscombe (1980), Wiscombe and Warren (1980) and Kokhanovsky and Zege (2004). In this work, we propose an analytical snow albedo and reflectance model, which can be used to derive near - surface snow optical and microphysical properties using measurements at just two to four wavelengths in the visible and near–infrared depending on the measurement system and type of snow. In particular, we present the method for the determination of snow grain size, absorption Angström coefficient and spectral absorption coefficient of impurities embedded in the snow matrix assuming an external mixture of snow grains and impurities. A technique to derive the snow albedo from reflectance measurements is also presented. The absorption and extinction of light by snow grains is treated in the framework of a geometrical optical approximation. The absorption coefficient of impurities is modeled using the Angström power law. All derivations

are performed in the framework of the asymptotic radiative transfer theory (see, e.g.,
Kokhanovsky and Zege, 2004,  Zege et al., 2011). It is assumed that the snow layer is vertically
and horizontally homogeneous and semi-infinite. Therefore, the effects of the finite layer
thickness are ignored.

**2.  Theory**
**2.1 The snow reflectance**
The snow reflectance $R$ (equal to unity  for ideal white Lambertian reflectors, see Appendix A)
can be presented in the following way using approximate asymptotic radiative transfer theory
(Kokhanovsky and Zege, 2004):

$$R = R_0 r_s^x,  \tag{1}$$

where  $x = u(\mu_0)u(\mu)/R_0$,  $R_0$  is the reflectance of a semi-infinite non-absorbing snow layer,
$u(\mu_0) = \dfrac{3}{7}(1+2\mu_0)$,  $\mu_0$  is the cosine of the solar zenith angle,  $\mu$ is the cosine of the viewing
zenith angle,  $r_s$  is the snow spherical albedo:

$$r_s = e^{-y},  \tag{2}$$

where

$$y = 4\sqrt{\dfrac{1-\omega_0}{3(1-g)}},  \tag{3}$$

$g$ is the asymmetry parameter, $\omega_0$ is the single scattering albedo. Let us introduce the probability
of photon absorption $\beta \equiv 1 - \omega_0$. It is equal as the ratio of absorption $\kappa_{abs}$ and extinction $\kappa_{ext}$
coefficients:
$$\beta = \frac{\kappa_{abs}}{\kappa_{ext}}, \tag{4}$$

where
$$\kappa_{abs} = \kappa_{abs}^{ice} + \kappa_{abs}^{pol}. \tag{5}$$

The first and second terms in Eq. (5) correspond to the ice grains and pollutants, respectively.
We assume that scattering and extinction of light by impurities is much smaller than that by ice
grains and, therefore (Kokhanovsky and Zege, 2004),
$$\kappa_{ext} = \frac{3c}{d}. \tag{6}$$

Here, $d = 1.5\bar{V}/\bar{S}$ is the effective diameter of ice grains, $\bar{V}$ is the average volume of grains, $\bar{S}$
is their average projected area averaged over all directions (equal to $\sum/4$ for convex particles in
random orientation, where $\sum$ is the average surface area and $c$ is the volumetric concentration of
the snow grains). The value of $c$ is equal to the volume of grains in unit volume of snow
($c = N\bar{V}$, where $N$ is the number of snow grains in unit volume of snow ($cm^{-3}$)). It is related to
the dry snow density $\rho_s$ by the following relation: $\rho_s = c\rho_i$, where $\rho_i$ is the bulk ice density.
The product of the effective diameter $d$ and the bulk ice absorption coefficient $\alpha$ is a small
number in the visible and near-infrared. Then it follows (Kokhanovsky and Zege, 2004, see their
Eq. (37) for the absorption path length inversely proportional to the absorption coefficient) that:
$$\kappa_{abs}^{ice} = B\alpha c, \tag{7}$$

where $B$ is the grain shape-dependent parameter (absorption enhancement parameter),
$\alpha = \dfrac{4\pi\chi}{\lambda}$, where $\chi$ is the imaginary part of the ice refractive index at the wavelength $\lambda$.

93        We present the absorption coefficient of pollutants in snow as

$$\kappa_{abs}^{pol}(\lambda) = \kappa_0 \tilde{\lambda}^{-m} , \tag{8}$$

where $\kappa_0 \equiv \kappa_{abs}^{pol}(\lambda_0)$, $\tilde{\lambda} = \lambda / \lambda_0$, $\lambda_0 = 1 \ \mu m$, $m$ is the absorption Angstrom coefficient.
It follows from Eqs. (4)-(8):
$$\beta = \frac{B\alpha d}{3} + \beta^{pol} , \tag{9}$$

where
$$\beta^{pol} = \frac{\kappa_0 \tilde{\lambda}^{-m} d}{3c} \tag{10}$$

and therefore:
$$y = \frac{4}{3}\sqrt{\frac{(B\alpha + \kappa_0 \tilde{\lambda}^{-m} c^{-1})d}{1-g}} . \tag{11}$$

Let the parameter $z = y^2$, from which it follows that:

$$z = (\alpha + f\tilde{\lambda}^{-m})l, \tag{12}$$

where

$$f = \frac{\kappa_0^*}{B}, \tag{13}$$

$\kappa_0^* = \kappa_0 / c$ and

$$l = \xi d \tag{14}$$

is the effective absorption length (EAL) and

$$\xi = \frac{16B}{9(1-g)} \tag{15}$$

is a grain shape (but not the grain size) dependent parameter.
The parameter $l$ can be determined directly from reflectance or albedo measurements, enabling
also the determination of the grain diameter $d = l / \xi$ assuming a particular shape of grains. It has
been found that the asymmetry parameter of crystalline clouds is usually in the range 0.74-0.76
in the visible (Garret, 2008). The asymmetry parameter $g$ for snow has not been measured so far
*in situ* but we shall assume that it is close to that in crystalline clouds and adopt the value 0.75. It
follows from experimental studies of Libois et al. (2014) that $B=1.6$ on average. Therefore, it
follows (see Eq. 15): $\xi \approx 11.38$.
Using the EAL, the equations for the snow reflectance and spherical albedo may be simplified.
Namely, it follows:

$$R = R_0 \exp(-x\sqrt{(\alpha + f\tilde{\lambda}^{-m})l}), \tag{16}$$

$$r_s = \exp(-\sqrt{(\alpha + f\tilde{\lambda}^{-m})l}). \tag{17}$$

The plane albedo can be derived as well (Kokhanovsky and Zege, 2004):
$$r = \exp(-u(\mu_0)\sqrt{(\alpha + f\tilde{\lambda}^{-m})l}).$$
(18)

The relationship between the albedo and the reflectance $R$ is given in Appendix A. It follows
from Eq. (16) that the spectral reflectance of polluted snow is determined by four *a priori*
unknown parameters: $l, R_0, f, m$. They can be estimated from the measurements of reflectance at
four wavelengths. This also enables the determination of the spectral reflectance (and albedo, see
Eq.(18)) at the visible and near – infrared wavelengths at an arbitrary $\lambda$. It follows:
$$R_1 = R_0 \exp(-x\sqrt{(\alpha_1 + f\tilde{\lambda}_1^{-m})l}),$$
(19)

$$R_2 = R_0 \exp(-x\sqrt{(\alpha_2 + f\tilde{\lambda}_2^{-m})l})$$
(20)

$$R_3 = R_0 \exp(-x\sqrt{(\alpha_3 + f\tilde{\lambda}_3^{-m})l})$$
(21)

$$R_4 = R_0 \exp(-x\sqrt{(\alpha_4 + f\tilde{\lambda}_4^{-m})l})$$
(22)

where the numbers 1, 2, 3, and 4 signify the wavelengths used. Equations (19)-(22) can be used
to compute four unknown parameters given above, and, therefore, determine reflectance and
albedo at any wavelength in the visible and the near-infrared using Eqs. (16)-(18). Let us assume
that the spectral channels are selected in a way that the effects of ice absorption can be neglected
in the first two channels $(\lambda_1, \lambda_2)$ and effects of absorption by pollutants are negligible in the
second pair of channels $(\lambda_3, \lambda_4)$. This situation is typical of not heavily polluted snow. Then it
follows instead of Eqs. (19)-(22):
$$R_1 = R_0 \exp(-x\sqrt{f\tilde{\lambda}_1^{-m}l}),$$
(23)

$$R_2 = R_0 \exp(-x\sqrt{f\tilde{\lambda}_2^{-m}l}),$$
(24)

$$R_3 = R_0 \exp(-x\sqrt{\alpha_3 l}),$$
(25)

$$R_4 = R_0 \exp(-x\sqrt{\alpha_4 l}).$$
(26)

Eqs. (25), (26) can be used to find the pair $(l, R_0)$:
$$R_0 = R_3^{\varepsilon_1} R_4^{\varepsilon_2}, \quad l = \frac{1}{x^2 \alpha_4} \ln^2\left[\frac{R_4}{R_0}\right],$$
(27)

where $\varepsilon_1 = 1/(1-b), \varepsilon_2 = 1/(1-b^{-1}), b = \sqrt{\alpha_3/\alpha_4}$. Then it follows from Eqs. (23), (24) that:
$$m = \frac{\ln(p_1/p_2)}{\ln(\lambda_2/\lambda_1)},$$
(28)

$$f = \frac{p_1 \tilde{\lambda}_1^m}{x^2 l},$$
(29)

where $p_k = \ln^2(R_k/R_0)$. In case of the absence of pollutants, Eqs. (27) remain valid. However,
the parameters $m$ and $f$ are undefined and $R = R_0 \exp(-x\sqrt{\alpha l})$.
One may also derive the impurity absorption coefficient at the wavelength
$\lambda_0$ normalized to the concentration of ice grains $c$ (see Eq. (13)):
$$\kappa_0^* = Af,$$
(30)

where $f$ is given by Eq.(29). The normalized absorption coefficient at each wavelength can also
be found using Eqs. (8), (28), (30).
To determine the concentration of pollutants $(c_p)$ one must either know in advance or determine
the impurity volumetric absorption coefficient defined as:
$$K(\lambda_0) = \frac{\bar{C}_{abs}(\lambda_0)}{\bar{V}},$$
(31)

where $\bar{C}_{abs}$ is the average absorption cross section of impurities and $\bar{V}$ is the averge volume of absorbing
impurities. Namely, it follows by definition:
$$c_p = \frac{\kappa_0}{K(\lambda_0)}$$
(32)

and

$$\mathbb{C} = \frac{\kappa_0^*}{K(\lambda_0)},$$
(33)

where $\mathbb{C} = c_p / c.$
The value of $K(\lambda_0)$ can be found, if one knows the type of pollutants and their microphysical properties.
In particular, it follows for the impurities much smaller than the wavelength $\lambda_0$ (van de Hulst, 1981) that :
$$K(\lambda_0) = F\alpha_{pol}(\lambda_0),$$
(34)

where
$$\alpha_{pol}(\lambda_0) = \frac{4\pi\chi_{pol}(\lambda_0)}{\lambda_0}$$
(35)

is the pollutant bulk absorption coefficient, $\chi_{pol}(\lambda_0)$ is the imaginary part of pollutant refractive
index and $n_{pol}$ is the real part of the pollutant refractive index,
$$F = \frac{9n_{pol}}{\left(n_{pol}^2 + 1 - \chi_{pol}^2\right)^2 + 4n_{pol}^2 \chi_{pol}^2} \quad . \tag{36}$$

It follows that $F = 0.9$ for soot (assuming that $n=1.75$, $\chi_{pol} = 0.47$ in the visible). One can see
that $\mathbb{C}$ can be found if one knows the refractive index of absorbing Rayleigh particles in
advance.

In particular, it follows for soot impurities that:

$$\mathbb{C} = \frac{Ap_1\tilde{\lambda}_1^m}{x^2 l F \alpha_{pol}(\lambda_0)}. \tag{37}$$

178          In case of non-Rayleigh scatterers, one needs to know not only the refractive index but

also the particle size distribution and shape of particles, enabling the determination of the
impurity volumetric absorption coefficient $K(\lambda_0)$ and, therefore, the normalized concentration
of impurities
$$\mathbb{C} = \frac{Ap_1\tilde{\lambda}_1^m}{x^2 l K(\lambda_0)}. \tag{38}$$



### 2.2. The snow albedo

### 2.2.1 Theory

If the plane albedo is the measured physical quantity one needs to find only three constants:

$l, f, m.$

The respective analytical equations can be presented as:

$$r_1 = \exp(-u(\mu_0)\sqrt{(\alpha_1 + f\tilde{\lambda}^{-m})l}),$$
(39)

$$r_2 = \exp(-u(\mu_0)\sqrt{(\alpha_2 + f\tilde{\lambda}_2^{-m})l}),$$
(40)

$$r_3 = \exp(-u(\mu_0)\sqrt{(\alpha_3 + f\tilde{\lambda}_3^{-m})l}).$$
(41)

We shall assume that the last channel is not influenced by impurities and the first two channels

are not influenced by the absorption of light by grains. Then it follows that:

$$r_1 = \exp(-u(\mu_0)\sqrt{f\tilde{\lambda}_1^{-m}l}),$$
(42)

$$r_2 = \exp(-u(\mu_0)\sqrt{f\tilde{\lambda}_2^{-m}l}),$$
(43)

$$r_3 = \exp(-u(\mu_0)\sqrt{\alpha_3 l}).$$
(44)

The EAL can be found from Eq. (44):

$$l = \frac{\ln^2 r_3}{u^2(\mu_0)\alpha_3}.$$
(45)

It follows from Eqs. (42), (43) that:
$$m = \frac{\ln(\psi_2/\psi_1)}{\ln(\lambda_1/\lambda_2)}, f = \frac{\psi_1 \tilde{\lambda}_1^m}{u^2(\mu_0)l}, \qquad (46)$$

where $\psi_k = \ln^2 r_k$.

203        In case of unpolluted snow, one derives:

$$r = \exp(-u(\mu_0)\sqrt{\alpha l}). \qquad (47)$$

Eq. (45) can be used to find the effective absorption length and, therefore, the spectral albedo of
unpolluted snow at any wavelength using Eq. (47). If not plane but rather spherical albedo is
measured, then all equations presented in this section are valid except one should assume that
$u = 1$ and substitute $r$ by $r_s$ (Kokhanovsky and Zege, 2004).
**3.  Experiment**

210        **3.1 The measurements of the plane albedo**

We have applied the technique developed above to the measured spectral plane albedo both for
polluted and pure snow. Therefore, in-situ spectral albedo measurements were obtained from two
different field sites located in the French Alps (polluted snow) and in Antarctica (clean snow).
*The spectral albedo of a spring alpine snowpack* was measured at the Col du Lautaret field site
(45°2' N, 6°2' E,  2100 m a.s.l.) in the French Alps. The measurements were performed using a
non-automated version of the spectrometer system described above. The hand-held instrument
has a single light collector, located at the end of 3 m boom placed 1.5 m above the surface. The
boom is rotated by the operator to successively acquire the downward and upward solar
radiation. The spectral albedo data (each spectral albedo measurement at a given point is an
average of five measurements)  at several locations close to the location Col du Lautaret field site
was obtained on 12th April,  2017 across a 100 m transect, in attempt to account for spatial
variability. The measurements were acquired in clear sky conditions, with a solar zenith angle
varying between 47.9º and 52.2º.
The results of comparison of measurements and the theory presented above are illustrated in
Fig.1 at the  Col du Lautaret field site. The parameters $l, f, m$ have been found from Eqs. (42)-
(44) and the measurements at the wavelengths $\lambda_1 = 400 nm, \lambda_2 = 560 nm, \lambda_3 = 1020 nm.$ At other
measurement sites across a transect the results of the inter-comparison are excellent and similar
to that presented in  Fig.1. Therefore, the theory can be used to derive snow optical and
microphysical properties even for polluted snowpack. The derived spectral probability of photon
absorption for the case shown in Fig. 1 is presented in Fig.2. The derived absorption coefficient
(assuming c=1/3), the grain diameter $d$ and the absorption Angström parameter $m$ for five sites
across the transect are listed in Table 1 (lines 1-5). It follows that the value of $m$ is in the range
2.4 - 4.1 consistent with the identified presence of dust particles in snow (Doherty et al., 2010).
The pure black carbon impurities have the values of $m$ close to one. The grain diameter is in the
range 1.7-2.2 mm consistent with low values of snow albedo at 1020nm (see Fig.1). Wiscombe
and Warren (1981) have calculated the dependence of the clean snow spectral albedo at the solar
zenith angle 60 degrees and several grain radii and presented it in their Fig.8. It follows from
their calculations that the albedo decreases from 0.8 to 0.4 while the diameter of grains changes
from 0.1 to 2mm. It follows from our Fig.1 that the measured plane albedo is close to 0.45
signifying the dominance of large grains in the snowpack as reported in Table 1.
*The spectral albedo of pure snow* (very low amount of impurities) was measured at Dome C
(75°5' S, 123°17' E), in Antarctica using an automated spectral radiometer (Libois et al., 2015;
Picard et al., 2016; Dumont et al., 2017). The instrument is composed of two individual heads
located approximately 1.5 m above the surface. Each head contains two cosine receptors facing
upward and downward, which receive the incident solar radiation and the reflected radiation. The
collectors are connected to a MAYA2000 PRO Ocean Optics spectrometer with fibre optics
through an optical switch. Radiation is measured over 350-2500 nm spectral range with an
effective spectral resolution of 3 nm.  Albedo was calculated as the ratio of the upward and
downward spectral irradiance. The full description of the instrument and the processing steps to
calculate the spectral albedo are given by Picard et al. (2016). The spectral albedo measurements
used here were made on the 10$^{th}$ January 2017, with a solar zenith angle of 63.2°, during clear
sky conditions assessed by ground observations.
The results of the application of the proposed technique to the pure snow (no pollution) albedo
measured in Antarctica are illustrated in Fig.3. Application of our technique results in excellent
agreement with measured albedo over pure snow (now pollution) in Antarctica. Because the
snow at Dome C is clean/pristine, the value of *f* is negligible, resulting in snow albedo depending
only on the effective absorption length/grain size, which has been derived at a single wavelength
(1020nm). The derived grain diameter for the case presented in Fig.3 is equal to 0.5mm. The
retrieval error estimation is presented in Appendix B.



## 3.2 The measurements of the spectral reflectance

The application of the developed theory to the measurements of the spectral reflectance is presented in Fig.4 for two locations with different dust loads (39.6ppm and 107.4ppm). The spectral reflectance of snow was measured in the European Alps (Artavaggio plains, 1650m a.s.l., 45°55'56.70"N; 9°31'33.28" E) at the solar zenith angle equal to 52 degrees. The measurements were made on March 14[th] 2014, after a major transport and deposition of mineral dust from the Saharan desert. The event was very intense, and it was reported in the recent scientific literature regarding snow optical properties, (Di Mauro et al., 2015; Dumont et al., 2017), atmospheric chemistry and physics (Belosi et al., 2017), and also microbiology (Weil et al., 2017). The dust transport event deposited fine mineral dust particles from the atmosphere via wet deposition, according to the BSC-DREAM-8b model (Basart et al., 2012). Spectral measurements of snow were made using a field spectrometer (Field Spec Pro, Analytical Spectral Devices, ASD). This instrument features a spectral range of 350-2500 nm, a full width at half maximum of 5–10 nm, and a spectral resolution of 1 nm. Data presented here were collected under clear sky conditions at noon. Incident radiation was estimated using a Lambertian Spectralon panel. Reflected radiance was divided by incident radiance, and the hemispherical conical reflectance factor was calculated for two plots containing 39.6 and 107.4 ppm of dust. Dust concentration was measured with a Coulter Counter by integrating particles with a diameter smaller than 18 μm. Spectral measurements were performed at nadir using a bare optical fiber (field of view of 25°) at 80 cm from the snow sample. Both the optical fiber and the spectralon panel were equipped with an optical level. Further details on this dataset can be found in Di Mauro et al. (2015).

One can see that the theory works well not only for the albedo measurements (see the
previous section) but also for the reflectance measurements for polluted snow layers. In
particular, our results are closer to the measurements as compared to the theoretical model
described by Flanner et al. (2007) (see Fig.4b in Di Mauro et al., 2015). The derived parameters
are given in Table 1 (lines 6-7). The value of m is *4.1* for the case with the 39.6ppm dust
concentration and it is *6.4* for the case with 107.4 dust concentration. Because the difference is
quite large for the close locations we conclude that snow also contained other pollutants (say,
soot) and the determined value of *m* represents the combined effect with larger values of *m* for
larger concentrations of dust, which is consistent with other observations of this parameter in
snow (Doherty et al., 2010). The retrieved absorption coefficient of snow pollutants (at the
wavelength $\lambda^* = 560nm$) is $0.1191\,m^{-1}$ for the dust concentration 39.6ppm and it is $0.3123\ m^{-1}$
for the dust concentration of 107.4 ppm. Assuming that the dust chemical composition and also
the dust particle size distribution are the same at both locations we can assume that the ratio of
absorption coefficients at two locations should be equal to the ratio of dust concentrations. The
difference between the two ratios is <3%, which is within the measurement uncertainty (10% for
dust load measurements), and suggesting that the retrieved absorption coefficients at the two
sites are consistent with each other.
The mass absorption coefficient (MAC) can be estimated using:

$$K_m = \frac{\kappa_{abs}^{pol}\left(\lambda^*\right)}{\mathbb{C}\rho c},$$

(48)


where $\rho$ is the density of the substance of impurities. Assuming that:

$\rho = 2.62g\,/\,cm^3$ (as for quartz), $c = 1/3$, $\mathbb{C} = 107.4\,ppm$ and $\kappa_{abs}^{pol}\left(\lambda^*\right) = 0.3123m^{-1}$,

(49)

one can derive that:

$$K_m = 0.0033 m^2/g, \qquad (50)$$

which is consistent with the values of MAC given by Utry et al.(2015) (e.g.,
$0.0023 m^2/g$ for quartz and $0.0051 m^2/g$ for illite (see their Table 1)).
**4. Conclusions**
In this work, we have presented a sequence of analytical equations, which can be used to
determine the snow grain size, the absorption coefficient of impurities, and the absorption
Angström coefficient of surface snow impurities from the snow reflectance measured at four
wavelengths: two in the visible and two in the near infrared as suggested by Warren (2013). In
the case of albedo measurements just three wavelengths can be used to find main snow
properties. For unpolluted snow, it is enough to perform the measurements at two wavelengths
(for reflectance measurements) or just at a single wavelength (for albedo measurements) in the
near-infrared to determine the snow grain size.
In principle, the refractive index of dust and dust size distribution can be also determined using
derived spectral absorption coefficient of dust and assuming the shape of dust particles.
However, we did not make an attempt for such retrievals in this work. The method for the
retrieval of the complex refractive index and single scattering optical properties of dust deposited
in mountain snow based on exact radiative transfer calculations has been proposed by McKenzie
Siles et al. (2016) in the assumption that local optical properties of dust grains can be simulated
assuming the spherical shape of particles. Their method is based on the extraction of dust grains
from snowpack. Our technique does not require such a complicated procedure.
We have demonstrated how snow albedo can be derived from spectral reflectance measurements
avoiding complicated integration with respect to the observation geometry (azimuth, viewing
angle). The last point is useful for the determination of the snow *albedo* from spectral *reflectance*
measurements (say, from aircraft or satellite) at a fixed observation geometry. Although the
comprehensive validation of the retrievals has not been attempted, we have found that the ratio
of derived absorption coefficients of pollutants at two concentrations is close to the ratio of
pollutant concentrations derived independently, which indeed should be the case taking the
proximity of two measurement sites with different dust loads. The general validity of the
approach is proven using field measurements (Alps, Antarctica) of both spectral reflectance and
plane albedo.
The determination of the EAL $l$ (unlike the effective grain diameter $d$) both from reflectance
and albedo measurements is practically insensitive to *apriori* unknown shape of ice crystals.
Therefore, this length may be useful for the characterization of snowpack microstructure (in
addition to the grain size $d$). The results presented in this work are useful for the interpretation of
snow properties using both reflectance spectroscopy (Hapke, 2005) and imaging spectrometry
(Dozier et al., 2009). It is assumed that the semi - infinite snow layer is vertically and
horizontally homogeneous. The effects of the snow layer finite thickness, close packing effects,
snow vertical inhomogeneity, possible internal mixture of pollutants in snow grains, and
underlying surface albedo (ice, soil, grass) are ignored.



## 350 Appendix A. Nomenclature

| Physical parameter | Notation | Units | Definition | Comments |
|---|---|---|---|---|
| Snow absorption coefficient | $\kappa_{abs}$ | $m^{-1}$ | $N\bar{C}_{abs}$ | $N$-number of snow grains in unit volume($m^{-3}$) $\bar{C}_{abs}$-average absorption cross section of grains ($m^2$) |
| Snow extinction coefficient | $\kappa_{ext}$ | $m^{-1}$ | $N\bar{C}_{ext}$ | $N$-number of snow grains in unit volume($m^{-3}$) $\bar{C}_{ext}$-average extinction cross section of grains ($m^2$) |
| Probability of photon absorption | $\beta$ | - | $\kappa_{abs}/\kappa_{ext}$ | $\beta \ll 1$ in the approximation studied |
| Snow single scattering albedo | $\omega_0$ | - | $1-\beta$ | close to 1 in the approximation studied |
| Snow asymmetry parameter | $g$ | - | $g = \dfrac{1}{2}\int_0^\pi p(\theta)\sin\theta\cos\theta d\theta$ | $\dfrac{1}{2}\int_0^\pi p(\theta)\sin\theta d\theta = 1$ $\theta$ is the scattering angle(equal to $\pi$ in the exact backward scattering direction) $p(\theta)$ is the conditional probability of photon scattering in a given direction specified by the angle $\theta$ (phase function) |
| Bulk ice absorption coefficient | $\alpha$ | $m^{-1}$ | $\dfrac{4\pi\chi(\lambda)}{\lambda}$ | $\chi(\lambda)$ - imaginary part of bulk ice refractive index $\lambda$ - wavelength |
| Volumetric absorption coefficient of pollutants | $K$ | $m^{-1}$ | $\bar{C}_{abs}^{pol}/\bar{V}_p$ | $\bar{C}_{abs}^{pol}$ -average absorption cross section of impurities in snow ($m^2$),$\bar{V}_p$ is their average volume($m^3$), $K$ is _proportional_ to the bulk absorption coefficient of impurities in case they are much larger as compared to |

| | | | | |
|---|---|---|---|---|
| | | | | the wavelength (and weakly absorbing) or much smaller as compared to the wavelength (so called Rayleigh scatterers). The coefficient of proportionality (absorption enhancement factor) depends on the shape of particles and real part of their complex refractive index. |
| Effective absorption length | $l$ | $m^{-1}$ | $\dfrac{\ln^2 r_s}{\alpha}$ (for clean dry snow) | $r_s = \exp(-\sqrt{\alpha l})$ $r_s$-spherical albedo $\alpha$-bulk ice absorption coefficient This definition holds for clean dry snow only The general definition for dry snow is given by Eq. (17) |
| Reflectance | $R(\mu_0,\mu,\varphi)$ | - | $I^{\uparrow}(\mu_0,\mu,\varphi)/I_{Lamb}^{\uparrow}(\mu_0)$ | Ratio of intensity of light reflected from a given snowpack to that of an ideal Lambertian surface with albedo 1.0 $(\mu_0,\mu,\varphi)$-cosine of the solar zenith angle, cosine of viewing zenith angle, and relative azimuth, respectively |
| Plane albedo | $r(\mu_0)$ | - | $2\int\limits_0^1 \bar{R}(\mu_0,\mu,\varphi)\mu d\mu$ | $\bar{R}=\dfrac{1}{2\pi}\int\limits_0^\pi R(\mu_0,\mu,\varphi)\varphi d\varphi$ - reflectance averaged with respect to the azimuth, black sky albedo |
| Spherical albedo | $r_s$ | - | $2\int\limits_0^1 r(\mu_0)\mu_0 d\mu_0$ | white sky albedo |
| Volumetric concentration of grains | $c$ | - | $N\bar{V}$ | $N$ – number concentration of grains, $\bar{V}$ -average volume of grains, $c$ - fraction of unit volume occupied by ice grains ( usually around 0.3) |
| Mass concentration of | $\rho_s$ | $gm^{-3}$ | $N\bar{m}$ | $N$ – number concentration of grains, |

| grains (snow density) | | | | $\bar{m} = \rho_i \bar{V}$ -average mass of grains, $\rho_i$ - bulk ice density, $\rho_s = \rho_i c$ (for dry snow) |
|---|---|---|---|---|
| Bulk ice density | $\rho_i$ | $gm^{-3}$ | - | $\rho_i = 916.7 kg/m^3$ at $0^\circ C$ |
| Bulk pollutant density | $\rho_p$ | $gm^{-3}$ | - | |
| Volumetric concentration of impurities | $c_p$ | - | $N_p \bar{V}_p$ | $N_p$ – number concentration of pollution particles, $\bar{V}_p$ -average volume of pollution particles, $c$ - fraction of unit volume occupied by impurities |
| Normalized volumetric concentration of impurities | $\mathbb{C}$ | - | $c_p / c$ | $\mathbb{C} = \dfrac{\rho_i}{\rho_s} c_p$ $\rho_i$ - bulk ice density, $\rho_s$ - snow density, $c_p$ - volumetric concentration of impurities |
| Effective diameter of grains | $d$ | $m$ | $\dfrac{3\bar{V}}{2\bar{S}}$ | equal to the diameter for the collection of spherical grains of the same size, $\bar{V}$ -average volume of grains, $\bar{S}$ -average cross section of grains (perpendicular to incident light beam), |








  **Appendix B. The retrieval error estimation**

**1.   The effective absorption length and diameter of grains**
Let us consider  the error budget for the retrieved snow parameters. To simplify, we assume that
the snow parameters are derived using albedo measurements.
The value of $l$ is determined from measurements just at a single wavelength in the framework of
the theory given above. It follows from Eq. (45):
$$\frac{\Delta l}{l} = K \frac{\Delta r_3}{r_3},$$
(B.1)

where
$$K = \frac{2}{\ln r_3}.$$
(B.2)

Therefore, the relative effective absorption length retrieval error is directly proportional to the
relative measurement error in the measured albedo. Larger values of $l$ correspond to the smaller
albedo. So one conclude that that the error of retrieval of larger values of EAL are generally
smaller (see Eq.(B.2)). For the cases, presented in Figs. 1 and 2, the wavelength 1020nm has
been used in the retrieval process and one finds that $K$ is equal to -2.5 and -5.8, respectively.
Assuming the measurement error of 3%, one derives that EAL is determined with the accuracy -
7.5 and -17.4 %, respectively with better accuracy for the observations in Alps, because the
albedo is lower there. Also we see that the overestimation of the albedo in the experiment leads
to the underestimation of the EAL and other way around in case measurements which
underestimate the snow albedo because $K$ is negative. Because $K$ generally decreases with the
wavelength one must use the largest wavelengths to have better accuracy (say, 1020nm instead
of 865nm). The wavelength should not be above 1200nm or so (depending on the snow grain
size) because the underlying theory valid for weakly absorbing media only. So strong ice
absorption bands must be avoided. The value of the snow grain size is proportional to the value
of *l*. Therefore, our conclusions are also valid for the derived snow grain size assuming that one
knows the parameter $\xi$ ( see Eq.15) exactly. However, this parameter is known with some error.
The uncertainty in the parameter $\xi = 16B/3(1-g)$ is difficult to access because we rely on a priori
value for the snow asymmetry parameter and the absorption enhancement parameter *B*. We use
the following values: *B=1.6, g=0.75*. The reported values of *g* for   for crystalline clouds do not
go above 0.8. Therefore, the module of the  absolute error in the parameter *1-g* is smaller than
0.05 (and the relative error is below 20%). The value of *B* is usually in the range: 1.4-1.8 and,
therefore, the absolute error in the parameter *B* is equal approximately to  $\pm 0.2$ and, therefore,
the relative error is $\pm 12.5\%$. It  follows that the absolute value of the maximal relative error in
the parameter $\xi$  is close to 24%. The error could be smaller in case the assumptions used in the
derivations are closer to the actual snow conditions. We find that the maximal relative error in
the derived grain diameter for the cases shown in Fig. 1, 2 is 25-30% depending on the snow
type, which is substantially larger as compared to the error in the estimation of EAL.




**2. The spectral absorption coefficient of pollutants**

399          Let us consider the error budget for the retrieved spectral absorption coefficient of

pollutants. The absolute error of the retrieved parameter $\Delta x$ is defined as
$$\Delta x = \sqrt{\sum_{j=1}^{J} \left[ \frac{\partial x}{\partial y_j} \right]^2 \Delta y_j^2} \ . \tag{B.3}$$

This error depends on $j$ measurement errors of reflectance/albedo at $j$-channel $\Delta y_j$ , where $J$   is
the number of channels used to retrieve the corresponding parameter assuming that there are no
forward model errors. It a difficult task to estimate the forward model error theoretically. It
depends on the specific type of snowpack.
It follows from Eqs. (46), (B.3) :
$$\frac{\Delta f}{f} = \sqrt{ \Upsilon_1^2 \left[ \frac{dr_1}{r_1} \right]^2 + \Upsilon_3^2 \left[ \frac{dr_3}{r_3} \right]^2 }, \quad \frac{\Delta m}{m} = \sqrt{ \Pi_1^2 \left[ \frac{dr_1}{r_1} \right]^2 + \Pi_2^2 \left[ \frac{dr_2}{r_2} \right]^2 }, \tag{B.4}$$

where

$$\Upsilon_n = \frac{2}{\ln r_n}, \ \Pi_n = \frac{2}{\ln r_n \ln \left( \psi_2 / \psi_1 \right)}. \tag{B.5}$$

We conclude that the errors of the pair ($f$, $m$) determination increase, if the selected wavelengths

at two channles in the visible   are too close and if the logarithm of albedo at selected channels is

close to unity (say, weak concentration of pollutants for the channels in the visible). The albedo

decreases for the cases with illumination closer to nadir. Therefore, to reduce errors one needs to

use the measurements with larger deviations of Sun from the horizon direction.


The absorption coefficient of pollutants is given by the following equation ( see Eqs. (8),


(13),(30)):


$$\kappa_{abs}^{pol} = Bcf \, \tilde{\lambda}^{-m}.$$

(B.6)


Therefore, one derives


$$\frac{\Delta \kappa_{abs}^{pol}}{\kappa_{abs}^{pol}} = \sqrt{\left[\frac{\Delta B}{B}\right]^2 + \left[\frac{\Delta f}{f}\right]^2 + \left[\frac{dc}{c}\right]^2 + \ln^2\left(\tilde{\lambda}\right)\left[\frac{\Delta m}{m}\right]^2}.$$

(B.7)


One concludes that the errors in the estimated snow volumetric concentration $c$ (snow density,


see Appendix A), the absorption enhancement coefficient $B$, and also pair $(f,m)$ must be


combined to estimate the total error in the retrieved spectral absorption coefficient of impurities


in snow. The errors are lower, if one is interested in the spectral absorption coefficient of


impurities normalized to its value at a specific wavelength defined as


$$\tilde{\kappa}_{abs}^{pol} \equiv \frac{\tilde{\kappa}_{abs}^{pol}\left(\lambda\right)}{\tilde{\kappa}_{abs}^{pol}\left(\lambda_*\right)},$$

(B.8)

where $\lambda_*$ is the selected wavelength (say, 550nm).

One derives for this coefficient:


$$\tilde{\kappa}_{abs}^{pol}\left(\lambda\right) \equiv \left(\frac{\lambda}{\lambda_*}\right)^{-m}$$

(B.9)

and, therefore, only the accuracy of the determination of absorption Angstroem exponent
influences the result:
$$\frac{\Delta \kappa_{abs}^{pol}}{\kappa_{abs}^{pol}} = \ln(\lambda / \lambda_*)^{-m} \frac{\Delta m}{m}.$$
(B.10)


An important point is the determination of concentration of pollutants in snow from optical

remote sensing data. In principle the concentration of pollutants can be found, if the absorption
coefficient of pollutants at a given wavelength is known. For instance,    it follows by definition
(see Eq. (31) for the definition of the volumetric absorption coefficient of impurities $K$):
$$c_p = \frac{\kappa_{abs}^{pol}(\lambda)}{K(\lambda)}.$$


Therefore, uncertainty in the derived or assumed value of $K$    (or mass extinction coefficient

$K_m = K / \rho_p$ , where $\rho_p$ is the density of the substance of a pollutant) influences the retrieval error
in addition to uncertainty of the derived absorption coefficient of pollutants $\kappa_{abs}^{pol}(\lambda)$. One can see
that the determination of the concentration of pollutants from optical remote sensing of
snowpack is a very challenging task.

In particular, one finds that the positive bias in the measured albedo in the visible will lead to

the underestimation of the concentration of pollutants (assuming that the grain size is exactly
known). It should be pointed out that in most cases the concentration of pollutants is so small
that it can not be assessed  using optical instruments (change in reflectance is inside experimental
measurement error). This issue has been discussed by  Zege et al. (2011) and Warren (2013).
Similar conclusions hold also if the reflectance (and not albedo) is the measured quantity.

**5.  Acknowledgments**
This work was mainly supported by the European Space Agency in the framework of ESRIN
contract   No. 4000118926/16/I-NB   ''Scientific Exploitation of Operational Missions (SEOM)
Sentinel-3 Snow (Sentinel-3 for Science, Land Study 1: Snow)''. CNRM/CEN and IGE are part of
labex OSUG@2020. Measurements in the French Alps were funded by the ANRJCJ grant EBONI
16-CE01-0006 and at Dome C by ANR JCJC MONISNOW 1-JS56-005-01.

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

Tables

Table 1. The derived snow parameters for the five samples. The value of $c$ is assumed to be equal
1/3, which leads to the extinction length ($l_{ext} = 1/\kappa_{ext}$) to be equal to the effective grain diameter
$d$. The absorption coefficient is given at the wavelengths $\lambda_0 = 1000\text{nm}$ and $\lambda^* = 560nm$.

| N | $\kappa_{abs}^{pol}(\lambda_0), m^{-1}$ | $\kappa_{abs}^{pol}(\lambda^*), m^{-1}$ | $m$ | $d, mm$ | Site |
|---|---|---|---|---|---|
| 1 | 0.0182 | 0.1954 | 4.1 | 2.1 | Col du Lautaret  (site 1) |
| 2 | 0.0342 | 0.2668 | 3.5 | 2.2 | Col du Lautaret  (site2) |
| 3 | 0.1073 | 0.7194 | 3.3 | 1.7 | Col du Lautaret (site 3) |
| 4 | 0.0769 | 0.5324 | 3.3 | 1.9 | Col du Lautaret (site 4) |
| 5 | 0.0943 | 0.3848 | 2.4 | 2.2 | Col du Lautaret (site 5) |
| 6 | 0.0111 | 0.1191 | 4.1 | 2.5 | Artavaggio plains (site 1) |
| 7 | 0.0077 | 0.3123 | 6.4 | 1.5 | Artavaggio plains (site 2) |





Figures

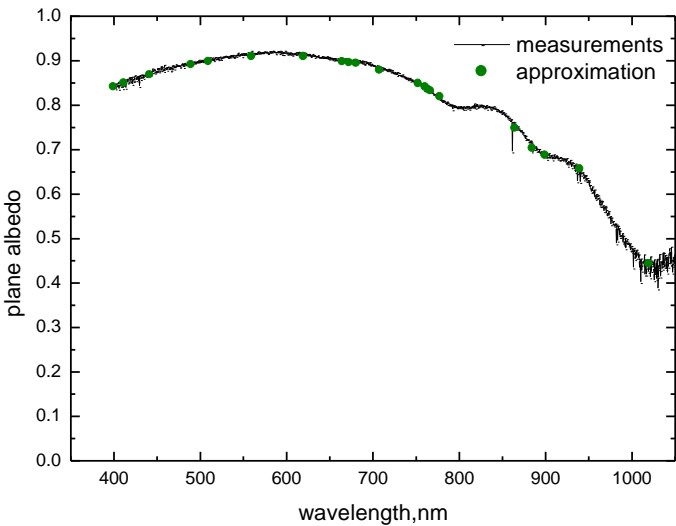


Fig.1. The intercomparison of theory (symbols) with experimental measurements of plane albedo
(line, no noise removed) performed in French Alps (45°2' N, 6°2' E, 2100 m a.s.l.) obtained on
12/04/2017. The plane albedo is an average of 5 measurements performed between 08h55 and
09h30 UTC for a polluted (by dust) snowpack. The solar zenith angle for the measurements was
between 47° and 49°. The noise of measurements has not been removed and clearly seen in the
near infrared portion of the spectrum.

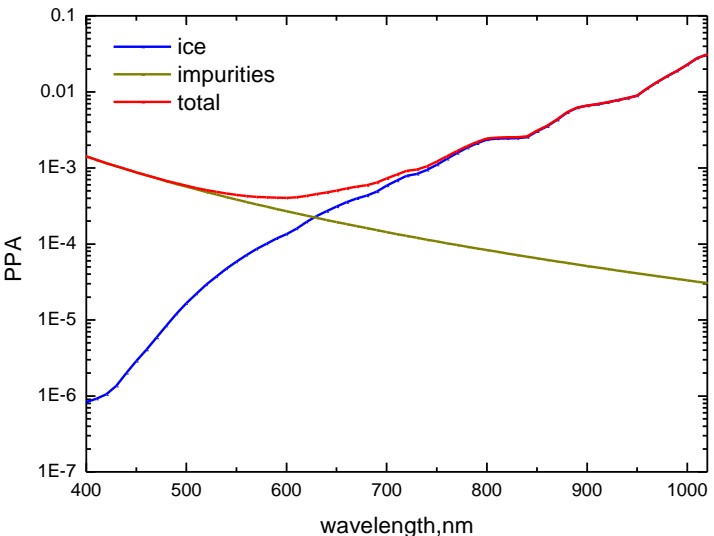


Fig.2. The derived spectral probability of photon absorption for the case presented in Fig.1.


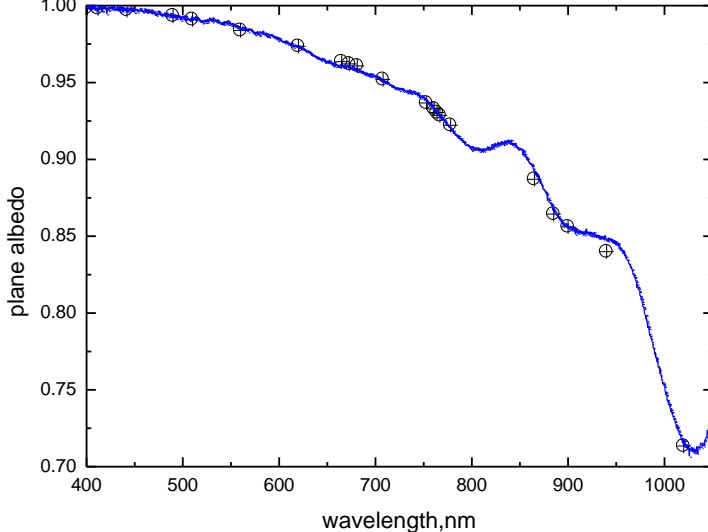


Fig.3 The inter-comparison of theory (symbols) with experimental measurements of plane albedo

(line) performed in Antarctica (Dome C, 75°5' S, 123°17' E) for pure snow. The measured plane

albedo was obtained on 10/01/2017 at 23h24 UTC, for a solar zenith angle of 63 degrees. The

parameters *l, f, m* have been derived from the measurements at 400, 560, and 1020nm.

568

569

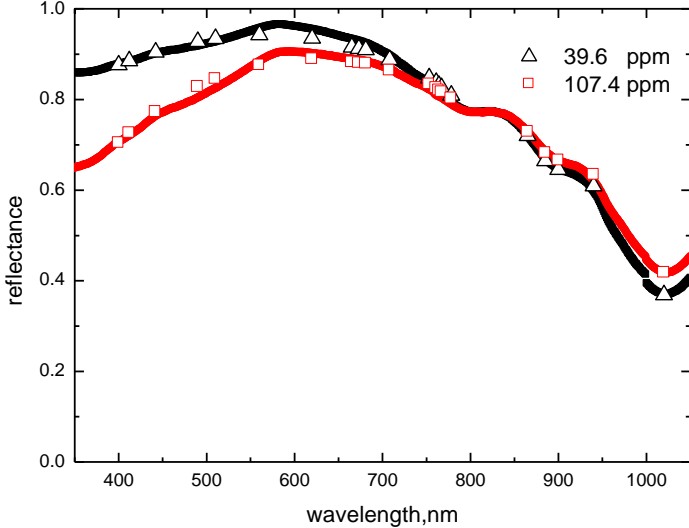

570

Fig.4 The inter-comparison of theory (symbols) with experimental measurements (line) in
European Alps (45°55'56.70"N; 9°31'33.28" E) for the polluted snowpack. The parameters $R_0$ , $l$,
$f$, $m$ have been derived from the measurements at 400, 560, 865 and 1020nm.
Reflectance measurements were collected on snow containing different concentration of dust:
39.6 ppm (black line) and 107.4 ppm (red line). The dust has been collected from the upper snow
layer ($\approx$5cm). Snow was clean at larger depths. A complete description of this dataset is
presented in Di Mauro et al. (2015).

578