# Peer review of "On the reflectance spectroscopy of snow"

_The Cryosphere, 2018_

## Referee Comment (RC1) · Anonymous Referee #1 · 14 May 2018

This study presents an impressive analytical technique for deriving snow grain size and impurity absorption coefficient from snow reflectance or albedo measurements. The technique is compared against a limited set of in-situ measurements of spectral albedo and reflectance, both of clean and highly contaminated snow, and is shown to work very well in reproducing the observed spectra and impurity contributions to absorption. Overall, this represents a nice demonstration of the analytical technique derived by Kokhanovsky and Nege (2004) and subsequent studies. My comments for improvement are relatively minor.

General comments:

1) Overall, the technique and its accuracy (against a small number of measurements) is very encouraging, but the main limitation I see with this analytical approach is its inabil-

ity to account for vertical heterogeneity in snow grain size and impurity content. This could be important in situations where grain size varies over the top several millimeters, and/or in situations where impurity content varies over the top several centimeters. In general, the sub-surface snow properties exert different relative contributions to the snow reflectance at different wavelengths. Thus in cases of substantial vertical variations in snow properties, a single grain size or impurity content will be insufficient for modeling the snow reflectance over the entire near-infrared and visible spectra, respectively. Given the nature of this study, I don't see this as a major problem, but I do think this limitation, along with the closely-related restriction of semi-infinite conditions, needs to be acknowledged more clearly, including in the abstract. A second limitation that should probably be acknowledged is the assumption that the spectral distribution of impurity absorption follows perfectly from the absorption Angstrom profile. This assumption could lead to issues in situations where the combination of impurities that are present in the snow do not abide by the Angstrom profile, as is the case with some biological constituents and types of dust.

2) The analytical derivations span 48 equations, and it is easy to become lost when going through these. To help ameliorate this, I suggest that the authors (a) include an appendix listing all of the terms in the equations, and (b) include fundamental SI units of the variables, where appropriate, both parenthetically in the text where they are introduced and also in the appendix. The latter will help readers to infer definitions of variables whose terminology varies across the radiative transfer literature. (For example, even the units of 'extinction coefficient' vary across texts).

3) Lines 194-207: This passage might be presented better as an expanded sub-section on the analytical relationships between errors in albedo/reflectance measurement and uncertainty in inferred quantities. An expanded discussion on the impacts of measurement error/uncertainty on inferred grain size and impurity absorption coefficient would be especially welcome here.

Minor comments:

line 44: Here I would explain that the technique is used to inter *near-surface* snow properties.

line 77: Here and elsewhere, please provide fundamental SI units of the quantity. (Is c the inverse of snow density?)

line 79: Please first define alpha before including this reference to it.

line 84: 'm' should be defined somewhere as the absorption Angstrom coefficient.

line 85: I don't recall sigma having been defined. I suspect this should be kappa.

lines 99-107: The asymmetry parameter (g) itself depends on particle shape, as has been shown in several studies (e.g., Yang and Liou, 1995, doi:10.1364/JOSAA.12.000162), but it appears here to be a constant that factors into the derivation of the grain shape dependent parameter (line 99). This seems confusing to me. Please devote a bit more discussion to the meaning of g in the context of its relationship to the shape parameter, and whether or not it needs to be assumed a constant in this framework.

line 143: Here it appears that kappa_0 only refers to absorption from impurities, but some of the earlier references to kappa indicate absorption by ice. This is an example where an appendix with all symbols and definitions would be helpful.

line 255 and Figure 4 mistakenly list the latitude twice without the longitude.

line 253: Over what depths of snow do these dust load measurements apply? This relates somewhat to general comment #1.

Figures: Please make the figures larger.

---

## Referee Comment (RC2) · Anonymous Referee #2 · 1 Jun 2018

Review of *'On the reflectance spectroscopy of snow'* by Kokhanovsky and 8 others

This manuscript describes an analytical method to derive snow grain size, the absorption Angstrom coefficient, the spectral absorption coefficient of impurities in snow, and the effective absorption length (EAL) from plane albedo and spectral reflectance measurements over snow. This approach is novel in its ability to model the ultra-violet to near-infrared hyperspectral reflectance of snow using between 3 and 4 snow reflectance or albedo measurements, given that at least one of the reflectance/albedo measurements is made in either the visible or near-infrared part for the spectrum. The manuscript provides a thorough step by step approach to extracting the important optical and microphysical properties of snow listed above but would benefit from editing to improve clarity and consistency with itself.

1.  Consider referencing, and then numbering, the figures in which their data is discussed in the text. Section 3 describes the different measurement types and data presented in the manuscript. Section 3.1 begins describing the measurements of spectral plane albedo made at Dome C, Antarctica. The plot of this data should be referenced here as Figure 1. The next part of section 3.1 describes the measurements of spectral plane albedo made at Col du Lautaret. The plot of this data should be referenced as Figure 2. The related discussion and derived spectral probability of photon absorption then becomes Figure 3. Finally, Section 3.2 discusses measurements of spectral reflectance made at a second site in the Alps. This data should remain, as it currently is, Figure 4.

2.  Table 1 is somewhat confusing. As is, the table presents two different types of data (plane spectral albedo and spectral reflectance), and data from two different locations (Col du Lautaret and a second location in the Alps). Additionally, the table caption only describes the first five rows of the table, which describe the plane spectral albedo measured at Col du Lautaret, and does not mention where the data included in rows 6-7 was measured. This lack of distinction can lead to confusion in the text itself, evidenced in line 244. Here, the authors are discussing the plane albedo measurements made at Col du Lautaret, which corresponds to rows 1-5 of Table 1. Line 244 gives the snow grain diameter range as 1.5 – 1.9 mm. However, the snow grain diameter range presented in rows 1-5 of Table 1 extends from 1.7 - 2.2 mm.

    At the least, I suggest including an additional row (or two) to the existing Table 1 to include the location and type measurements recorded in a specific row. Alternatively, splitting the data presented in Table 1 into two separate tables by data type and location to avoid confusion. A new Table 1 (Table1*) could present the plane albedo measurements made at Col du Lautaret (Table 1; rows 1-5), and a second table (Table 2*) could present the spectral reflectance measurements made at the second site in the Alps (Table 1; rows 6-7).

    Additionally, I suggest changing the notation of parameter description at the top of column 2 in Table 1 to match that of column 3. If I understand the table correctly, column 2 corresponds to the derived snow absorption coefficient for wavelength = 1000nm, while column 3 corresponds to the derived snow absorption coefficient for wavelength = 560 nm. The parameter notation at the top of column 3 clearly indicates this, while the parameter notation at the top of column 2 seems to suggest a completely different parameter is being described. If my above understanding is correct, I suggest changing the parameter description at the top of column 2 to match the form of column 3 and the table caption, i.e.:

    $$\kappa_{abs}^{pol}(\lambda_0)$$

Line 42:    "one to four wavelengths" $\rightarrow$ Discussion later in the manuscript refers to using a minimum of 3 wavelength measurements to derive $l$, $m$, and $f$ from plane albedo measurements (see lines 174-175), and 4 wavelengths measurements to derive $R_o$, $l$, $m$, and $f$ from snow reflectance (see lines 114-116). I don't believe the authors discuss using only 1 or 2 wavelength measurements to derive snow optical/microphysical properties anywhere in the paper. If it is possible to derive

snow properties from 1-2 wavelength measurements, the authors should elaborate on this, or adjust this statement in the introduction to reflect the need for 3-4 wavelength measurements.

Line 79: It's not clear to me how equation 7 follows from "$\alpha d \rightarrow 0$." The authors reference Kokhanovsky and Zege (2004), which presents the following definition for $\alpha$:

$$\alpha = 4\sqrt{\frac{\ell_{tr}}{3\ell_{abs}}}$$

where $\ell_{tr}$ and $\ell_{abs}$ are the transport path length and absorption path length respectively. This is different from the definition for $\alpha$ given on line 81 of this manuscript. Further expansion on what is intended by "$\alpha d \rightarrow 0$" in the context of equation 7 would be helpful here.

Line 84: Identifying the variable "$m$" in the vicinity of this equation would be helpful.

Line 85: "$\sigma_{abs}^{pol}$" is used in the definition of $\kappa_0$, but is not explicitly defined itself. I.e., is this referencing the absorption cross-section for pollutants in the snow?

Line 108: Suggest rephrasing the beginning to the sentence. Perhaps, "Using the EAL, the equations for snow reflectance…".

Line 115: I don't believe "$m$" has been explicitly defined up to this point in the manuscript. If not back at equation 8 (see comment above), here would be a great place to do so.

Line 174: Delete "In case" and begin the sentence with "If the plane albedo…"

Line 194: It is not clear how the authors arrive at equation 45 from equation 48. Additional explanation would be helpful.

Line 204: "(Assuming that the grain size is exactly known)." – Generally, snow grain size is not known "exactly." Given this, the parenthetical aside should be deleted, leaving the remainder of the sentence ("…one finds that the positive bias in the measured albedo in the visible will lead to the underestimation of the concentration of pollutants.") still valid.

Line 234: For consistency with the language used in the last sentence of the previous paragraph, suggest changing "inter-comparison" → "comparison"

Line 236-
237: My understanding is that 5 different spectral albedo measurements were taken and subsequently averaged to create the solid line in Figure 1. Line 234 seems to state that this average is what is shown and compared with the theory estimates in Figure 1. However, the sentence staring on line 236 and ending on 237 "At all measurement sites…presented in Fig. 1," makes it sound like measurements were taken at more than one location, even though the previous paragraph lists only one site (Col du Lautaret). Is the intent of this sentence to convey that the 5 independent albedo measurements taken at Col du Lautaret are similar to/consistent with their average? If so, I would suggest removing references to "sites" (e.g., Albedo measurements at all 5 points along the 100 m transect are similar to their averaged spectra show in Figure 1…).

Line 238: Consider combining with the previous sentence along the lines of: "All 5 albedo measurements are consistent with the average shown in Fig.1, supporting our assertion that the theory presented here can be used to derive snow optical/microphysical properties for a polluted snow pack."

Line 242: If "*m*" as used here is referring to the absorption Angstrom parameter, it should again be defined as such earlier in the manuscript.

Line 244: If I understand this paragraph correctly, the authors are referring to rows 1-5 in Table 1. While line 244 lists the grain diameters as 1.5 – 1.9 mm, the grain size diameter range given in rows 1-5 of Table1 is: 1.7 – 2.2 mm.

Additionally, this sentence could benefit from a reference as to why the low snow albedo values at 1020 nm are consistent with snow grain diameters 1.7 – 2.2 mm. There is a local minimum in hyperspectral snow albedo around 1020 nm (Wiscombe and Warren, 1980; Warren, 1982; Nolin and Dozier, 1993; Grenfell et al., 1994), but it varies by grain size. Citation of how the grain sizes discussed in this manuscript are consistent with these observations would help bolster the stated expectation of consistency.

Line 245-
248: These two sentences seem a little cumbersome and could benefit from some rephrasing. Perhaps something like: "Application of our technique results in excellent agreement with measured albedo over pure snow (no pollution) in Antarctica. Because the snow at Dome C is clean/pristine, the absorption Angstrom parameter (*m*) and *f* are negligible, resulting in snow albedo depending only on *l* (characteristic length)."

Line 247-
248: The authors refer to the "characteristic length." Is this a reference to the EAL defined in equation 14? If it is not, this is a new topic that I do not think has been addressed earlier in the paper and should be if it has not.

Line 267: This paragraph is discussing the measurements of spectral reflectance presented in Figure 4. It's unclear why the HCRF was calculated for dust concentration of 0.92 ppm, since spectral reflectance is only presented for dust concentrations of 39.6 ppm and 107.4 ppm. I would recommend either explaining why the calculations for 0.92 ppm are mentioned here, or just removing it entirely as its inclusion suggests there is addition data not presented in the manuscript.

Line 276: Suggest replacing "first case" and "second case" with "39.6 ppm dust concentration" and "107.4 ppm dust concentration" respectively for clarity.

Line 285-
286: This sentence is a little confusing as is. Suggest changing to something like: "The difference between the two ratios is < 3%, which is within the measurement uncertainty, and suggesting that the absorption coefficients at the two sites are consistent with each other."

It would also be helpful to state what the measurement uncertainty for the dust concentrations.

Line 293: Define the acronym "MAC" at line 286. (i.e., "The mass absorption coefficient (MAC)…")

Line 299-
301: Suggest editing the end of this sentence for clarity → "…snow reflectance measured at four wavelengths: two in the visible and two in the near-infrared as suggested by Warren (2013)."

Line 303-
308: Since the authors state in the previous sentence that they do not attempt to determine the refractive index of dust in this manuscript, the sentences starting at line 303 and ending at 308 ("The method for the retrieval of complex refractive index…. complicated procedure.") are unnecessary and should be removed.

Line 319: An acronym for effective absorption length, EAL, is presented on line 98. I would suggest either using throughout the paper, or don't define the acronym.

It may also be useful to list the EAL in lines 297-299 as one of the parameters derived from the analytical equations.

Figure 3: Include a legend in this figure as was done for the other figures.

Figure 3: Y-axis range should be 0 to 1 as in Figure 1.

Figure 4: Only latitudes are given in the figure caption. Add the longitudes as was done in the captions for Figures 1 & 3.

ALL
Figures: Set x-axis to have the same range across all figures (e.g., 400-1050nm).

References

Wiscombe, W. J., & Warren, S. G. (1980). A model for the spectral albedo of snow. I: Pure snow. *Journal of the Atmospheric Sciences*, *37*(12), 2712-2733.

Warren, S. G. (1982). Optical properties of snow. *Reviews of Geophysics*, *20*(1), 67-89.

Nolin, A. W., & Dozier, J. (1993). Estimating snow grain size using AVIRIS data. *Remote sensing of environment*, *44*(2-3), 231-238.

Grenfell, T. C., Warren, S. G., & Mullen, P. C. (1994). Reflection of solar radiation by the Antarctic snow surface at ultraviolet, visible, and near-infrared wavelengths. *Journal of Geophysical Research: Atmospheres*, *99*(D9), 18669-18684.

---

## Author Response (AR1)

_**Answer to a Reviewer 1:**_

The authors thank the reviewer for the constructive comments on the paper. Our answers are given below:

1. We acknowledge that the theory described must be extended to account for the possible snow vertical inhomogeneity and possible finite thickness of a snowpack. These topics are out of scope of this paper. The abstract, introduction, and conclusions have been modified to account for your comment.
2. We also agree that the retrieval approach will not work well in case of polluted snow with the spectral absorption coefficients of pollutants, which do not follow the Angström law. The abstract, introduction, and conclusions have been modified to account for your comment. Of course, general equations (Eqs. 1-3) to solve the direct problem of snow optics presented in the paper can be used anyway. Eq. 1 has a misprint (R0 is missed before the exponential term). We have corrected this misprint in the final version.
3. All equations and definitions are explained in the text. We have also prepared the Appendix A with all definitions and units. Also we have prepared a special section (appendix B) on discussion of the retrieval errors as advised by you.
4. With respect to your comment related to the asymmetry parameter $g$, we confirm that this parameter depends on the shape of the grains. We have used the fixed value (0.75) in the determination of the grain size. Although the effective absorption length $l$ (see Eq.14) can be derived from reflectance even if the value of $g$ is not known. Therefore, we propose to use $l$ for the characterization of natural snowpacks.
5. We have accounted for all your minor comments.

_**Answer to a Reviewer 2:**_

The authors thank the reviewer for the constructive comments on the paper. Our answers are given below:

1.      We have changed the sequence of the discussions of the measurements at different sites. Therefore, the numbering of figures has not been changed. The data shown in Figure 2 is related to Figure 1.  So we prefer the sequence of figures as it stands.

2.      Table 1 has been modified according your advices.

Line 42: We have mentioned in the conclusions that one can use just 1 wavelength to find the snow grain size in case of albedo measurements for clean snow. In case of reflectance measurements one needs to perform the measurements at two wavelengths (for clean snow).

Line 79: We have added an explanation related to Eq. 7 in the text.

Lines 84, 85, 108, 115, 174, 234, 242, 244,245-248, 267,276, 285-286, 293, 299-301, Figure 4: Done.

Lines 194, 204: We have prepared Appendix B to explain the derivations.

Lines 236-238: We had the measurements at 5 sites. Each measurement at each site has been performed 5 times and the average spectral curve has been found for all five sites. The text has been modified.

Lines 303-308: We agree that we do not retrieve refractive index of dust in this work. However, the extension of this work may lead to such retrievals. So we prefer to keep the text as it stands.

Line 319: We have modified the text taking into account your comment. Although we do not provide EAL in the Table. It can be calculated via the grain diameter as described in the paper.

Figure 3: The figure is clear and we do not think that the legend is really needed to clarify the discussion. Also we do not see a point in changing the scale.

All figures: We do not change the axis OX because different instruments operate in different spectral ranges.

[revised manuscript text omitted]

where

$$f = \frac{\kappa_0^*}{B}, \qquad (13)$$

$\kappa_0^* = \kappa_0 / c$ and

$$l = \xi d \qquad (14)$$

is the effective absorption length (EAL) and

$$\xi = \frac{16B}{9(1-g)} \qquad (15)$$

is a grain shape (but not the grain size) dependent parameter.

The parameter $l$ can be determined directly from reflectance or albedo measurements, enabling also the determination of the grain diameter $d = l / \xi$ assuming a particular shape of grains. It has been found that the asymmetry parameter of crystalline clouds is usually in the range 0.74-0.76

in the visible (Garret, 2008). The asymmetry parameter $g$ for snow has not been measured so far

*in situ* but we shall assume that it is close to that in crystalline clouds and adopt the value 0.75. It follows from experimental studies of Libois et al. (2014) that *B=1.6* on average. Therefore, it follows (see Eq. 15): $\xi \approx 11.38$.

Using the EAL, the equations for the snow reflectance and spherical albedo may be simplified.

Namely, it follows:

$$R = R_0 \exp(-x\sqrt{(\alpha + f\lambda/\sigma^m)l}), \qquad (16)$$

$$r_s = \exp(-\sqrt{(\alpha + f\lambda/\sigma^m)l}). \qquad (17)$$

The plane albedo can be derived as well (Kokhanovsky and Zege, 2004):

$$\qquad r = \exp(-u(\mu_0)\sqrt{(\alpha + f\lambda^{-m})l}). \tag{18}$$

The relationship between the albedo and the reflectance $R$ is given in Appendix A. It follows from Eq. (16) that the spectral reflectance of polluted snow is determined by four *a priori*

unknown parameters: $l, R_0, f, m$. They can be estimated from the measurements of reflectance at four wavelengths. This also enables the determination of the spectral reflectance (and albedo, see

Eq.(18)) at the visible and near – infrared wavelengths at an arbitrary $\lambda$. It follows:

$$\qquad R_1 = R_0 \exp(-x\sqrt{(\alpha_1 + f\lambda_1^{-m})l}), \tag{19}$$

$$\qquad R_2 = R_0 \exp(-x\sqrt{(\alpha_2 + f\lambda_2^{-m})l}) \tag{20}$$

$$\qquad R_3 = R_0 \exp(-x\sqrt{(\alpha_3 + f\lambda_3^{-m})l}) \tag{21}$$

$$\qquad R_4 = R_0 \exp(-x\sqrt{(\alpha_4 + f\lambda_4^{-m})l}) \tag{22}$$

where the numbers 1, 2, 3, and 4 signify the wavelengths used. Equations (19)-(22) can be used to compute four unknown parameters given above, and, therefore, determine reflectance and albedo at any wavelength in the visible and the near-infrared using Eqs. (16)-(18). Let us assume that the spectral channels are selected in a way that the effects of ice absorption can be neglected in the first two channels $(\lambda_1, \lambda_2)$ and effects of absorption by pollutants are negligible in the second pair of channels $(\lambda_3, \lambda_4)$. This situation is typical of not heavily polluted snow. Then it follows instead of Eqs. (19)-(22):

$$\qquad R_1 = R_0 \exp(-x\sqrt{f\lambda_1^{-m}l}), \tag{23}$$

$$R_2 = R_0 \exp(-x\sqrt{f \lambda_2^{-m} l}), \qquad (24)$$

$$R_3 = R_0 \exp(-x\sqrt{\alpha_3 l}), \qquad (25)$$

$$R_4 = R_0 \exp(-x\sqrt{\alpha_4 l}). \qquad (26)$$

[revised manuscript text omitted]

---

## Author Response (AR2)

*Dear Editor:*

*We have accounted for all your minor comments (see the marked manuscript). The terms of the form lambda^{%m} in fraction numerators are showing up correctly now.*

*Best wishes,*

*Alexander Kokhanovsky*

**Editor Decision: Publish subject to minor revisions (review by editor)** (03 Jul 2018) by Benjamin Smith
Comments to the Author:
The authors have addressed the comments from the referees, and the manuscript seems to be in good shape, except that the new text in the appendices needs some quick editing. I would also ask the authors to look carefully at how the equations have rendered in the PDF version of the article. The terms of the form lambda^{%m} in fraction numerators are not showing up correctly, which may require either modification to the authors' source files or consultation with the publication department at TC.

338: should be "the a priori"
369: should be "one would conclude" or "one should conclude"
370, delete comma after 'cases'
372: "with the accuracy of" should be "with accuracies of "
373: add comma after 'respectively'
375: add comma after EAL, "in case" should be "in the case of"
386: the term "module" is not familiar to me. Do you mean "modulus"? That doesn't entirely make sense to me. Maybe magnitude?
414: "use the measurements" should be "use measurements"
432: "determination of concentration" should be "determination of the concentration"